# Changes in 25-(OH) Vitamin D Levels during the SARS-CoV-2 Outbreak: Lockdown-Related Effects and First-to-Second Wave Difference—An Observational Study from Northern Italy

**DOI:** 10.3390/biology10030237

**Published:** 2021-03-19

**Authors:** Davide Ferrari, Massimo Locatelli, Martina Faraldi, Giovanni Lombardi

**Affiliations:** 1SCVSA Department, University of Parma, 43121 Parma, Italy; davide.ferrari@unipr.it; 2Laboratory Medicine Service, San Raffaele Hospital, 20132 Milano, Italy; locatelli.massimo@hsr.it; 3Laboratory of Experimental Biochemistry & Molecular Biology, IRCCS Istituto Ortopedico Galeazzi, 20161 Milano, Italy; martina.faraldi@grupposandonato.it; 4Department of Athletics, Strength and Conditioning, Poznań University of Physical Education, 61-871 Poznań, Poland

**Keywords:** 25-hydroxy vitamin D, SARS-CoV-2 infection, lockdown, sun exposure, age

## Abstract

**Simple Summary:**

Several biological functions, more or less sustained by experimental evidence, have been proposed for vitamin D and in addition to its role in bone health, its optimal status has also been related with a reduced risk of allergy, obesity, and respiratory tract infections. During the SARS-CoV-2 pandemic, vitamin D levels have been put into relation with both the susceptibility to SARS-CoV-2 infection and the severity of COVID-19. The results from different studies are, however, not conclusive, since several variables impact on this relationship. In this study, we investigated the impact of the stringent confinement during the March–April 2020 lockdown on vitamin D levels and the relationship with the recorded sun exposure in the considered area (Milano, Italy). Furthermore, we investigated the eventual relationship between vitamin D levels and SARS-CoV-2 infection in different age groups throughout the pandemic, including the first and second wave, as well as the lockdown and between-lockdown periods. Taken together, our data suggest that 25-(OH)D levels are neither related with sun exposure nor with SARS-CoV-2 infection.

**Abstract:**

Background. We investigated the changes in 25-hydroxy vitamin D [25-(OH)D] concentrations values, during the first and the second pandemic waves and the impact of the lockdown periods, with their different approaches to home confinement, among different age groups. Methods. Daily cloud-modified vitamin D UV dose (UVDVC), for the area of interest (Milano, Italy), was obtained. Single-center 25-(OH)D determinations from 40,996 individuals in a 2019 (pre-pandemic), 32,355 individuals from 1 January to 31 August 2020 (containing the first pandemic wave) and 27,684 individuals from 1 June to 30 November 2020 (containing the second wave) were compared to investigate both the effect of the lockdown on vitamin D status and the association between 25-(OH)D and SARS-CoV-2 positivity. Results. No direct association was found between UVDVC, 25-(OH)D status and SARS-CoV-2 infection. The stringent confinement implemented during the first wave lockdown has not had any effect on 25-(OH)D status, although some peculiar time-restricted profile behaviors can be deduced, possibly due to vitamin D supplementation habits and features of those patients who presented to the hospital during the lockdown period. Conclusions. Although our data do not suggest any direct relationship between 25-(OH)D status, sun exposure, and SARS-CoV-2 infection, an indirect protective role cannot be excluded.

## 1. Introduction

Beginning in April 2020, the hypothesis of a role for vitamin D in severe acute respiratory syndrome coronavirus 2 (SARS-CoV-2) infection risk and the related coronavirus disease (COVID-19) outcomes have been proposed [1].

As very recently reviewed, in addition to its classical hormone action related to calcium homeostasis and bone metabolism, vitamin D is supposed to be more or less directly involved in a plethora of physiological mechanisms aimed at maintaining the whole-body homeostasis. For example, in recent years, vitamin D has emerged as a potential immune modulator [2]. The most active metabolite of the vitamers D, 1α,25-(OH)_2_D [3], regulates development, maturation, and functioning of different lineages of immune cells [4]. Vitamin D, indeed, limits the activation of T lymphocytes and determine a phenotypic shift from T helper (Th)1/Th17 pro-inflammatory to the Th2/T regulatory (Treg) anti-inflammatory phenotype [5,6,7]. Antigen presenting cells (APCs, such as macrophages and dendritic cells) and epithelial barrier cells, which are involved in the identification of pathogens, express the 1α-hydroxylase which is responsible of the biosynthesis of 1α,25-(OH)_2_D. Cytochrome P 27B1 (CYP27B1), that holds the 1α-hydroxylase activity, is directly induced by pathogens throughout the activation of toll-like receptors (TLRs) by pathogen-associated molecular patterns (PAMPs). Instead, systemic calcitriol activation takes place at the kidney level and is under the control of calcium, parathyroid hormone (PTH) and fibroblast-growth factor (FGF)-23 [8,9]. Furthermore, 1α,25-(OH)_2_D stimulates the synthesis anti-microbial compounds (i.e., cathelicidins, defensins) [8], which are also active against viruses [10,11], reduce the availability of iron and the needed for pathogens replication in infected cells by inhibiting hepcidin expression [12]. Furthermore, 1α,25(OH)_2_D improves the epithelial barrier function at the gut [13,14] and the lung level [15], which are the main sites of entrance for pathogens, enhances neutrophils function [16] and stimulates the generation of reactive oxygen species (ROS) [17], that cooperate in killing pathogens, and stimulates the phagocytic and autophagocytic activity of macrophages [18]. Thereby, hypovitaminosis D has been proposed as an increased susceptibility factor for respiratory and systemic infections, although definitive clinical evidence is lacking [19,20].

Several reports have evidenced the association and sometimes, a causal link between hypovitaminosis D status and both the risk of SARS-CoV-2 infection and the severity and mortality of COVID-19 [2]. However, since the different approaches applied and the great differences in the cohorts studied, these associations have not always been confirmed [21]. Moreover, although several clinical trials are ongoing, preliminary but contrasting evidence is emerging concerning the usefulness of vitamin D supplementation on the clinics of COVID-19 patients [2].

Serum 25-(OH)D (D3 and D2) is considered the best indicator of vitamin D status [22,23] although definitive reference ranges for 25-(OH)D are still object of discussion [2] and different national institutions and scientific societies defined different reference ranges. The sufficiency threshold has been settled at 50 nmol/L by the US Institute of Medicine (IOM) and at 75 nmol/L by the Endocrine Society [24,25].

Only Australia and New Zealand apply variable reference ranges depending on the season, considering the dependence of the endogenous synthesis of vitamin D on sun exposure [26]. This is of particular relevance since, as well known, the endogenous biosynthesis of vitamin D is triggered in the skin by ultraviolet (UV) B radiation. However, as we recently demonstrated on a large mid-latitude north Italian cohort of 30,400 subjects over 13 years, UV exposure alone is not sufficient to explain the vitamin D status of the population and several other factors, including lifestyle habits and personal characteristics, are important variables [27]. Only a few studies have investigated the association between UVB exposure, 25-(OH)D levels and SARS-CoV-2 infection fate. However, all these investigations have been performed considering the first pandemic wave and consequently, since temporarily limited (more or less correspondent to the “vitamin D winter”) to a specific period of the year, cannot be definitively exhaustive in terms of association with vitamin D potential. Furthermore, the strict lockdown and the consequent home-confinement and possibly the further limitation of sun exposure may have had a relevant effect on vitamin D biosynthesis.

Based on this background, this study is aimed at evidencing the changes in 25-(OH)D concentrations along the pandemic period, including both the first and the second waves, and the impact of the lockdown periods, with their different approaches to home-confinement among different age groups.

## 2. Materials and Methods

### 2.1. UV and Ozone Data

Time-series UV dose data were obtained from the Tropospheric Emission Monitoring Internet Service (TEMIS), formerly part of the Data User Program (DUP) of the European Space Agency (ESA). The satellite data, available at http://www.temis.nl/uvradiation/ (accessed on 18 December 2020) in the form of HDF-4 files, additionally provide information about tropospheric trace gases and aerosol concentrations, cloud coverage, and surface albedo. UVR data used in this study were based on the global ozone fields measured by the SCIAMACHY instrument aboard the ENVISAT satellite [28]. Among the TEMIS locations, the Ispra (Varese, Italy) station (latitude 45°81′, longitude 8°63′), was chosen since it is located 50 km northeast of Milan (latitude 45°28′, longitude 9°10′). For this location, the cloud-modified vitamin D UV dose (UVDVC) data were retrieved.

### 2.2. Clinical Data

For the lockdown effect, serum total 25-(OH) vitamin D were evaluated at San Raffaele Hospital in Milan, Italy, in 40,996 individuals (13,360 males aged 55.3 ± 19.7 and 27,636 females aged 58.3 ± 18.3), who had their blood tested between 1 January and 31 August 2019, and 32,355 individuals (11,252 males aged 56.4 ± 19.6 and 21,103 females aged 58.7 ± 18.5) who had their blood tested between 1 January and 31 August 2020 (“first wave group”). For the association between 25-(OH)D and SARS-CoV-2 positivity during the second wave, serum total 25-(OH)D levels were evaluated in 27,684 individuals (9164 males aged 56.6 ± 19.6 and 18,520 females aged 59.0 ± 18.3) who had their blood tested between 1 June and 30 November 2020 (“second wave group”). According to a Fisher’s test, the three groups (2019, first wave, and second wave groups) were significantly different for the number of males and females. For this reason they were further stratified by gender and age.

The majority of COVID-19 cases during the first wave (first wave peak) were framed approximately between the 1 March and 30 April, whereas the second wave peak was framed approximately between 1 October and 30 November. Of the total of vitamin D tests, 95% were routine analysis while only 5% were from hospitalized patients. Blood samples were collected as previously described [29], and 25-(OH)D was assayed on a Roche COBAS 8000 (Roche, Basel, Switzerland) by a electrochemiluminescence immunoassay. According to manufacturers’ datasheet (ref. 07028148500V3.0) the inter-assay and intra-assay coefficients of variation (CV) for 25-(OH)D measurements were between 1.1 and 2.4% for the higher concentration standards (83.4 and 93.6 ng/mL) and 3.1 and 10.8% for the lower concentrations standards (9.83 and 29.2 ng/mL).

To assay the presence of SARS-CoV-2 RNA, RT-PCR was performed on samples from nasopharyngeal swabs on a Roche-Cobas Z480 thermocycler (Roche, Basel, Switzerland). All subjects involved in the study gave their informed consent to the anonymous use of their data for retrospective observational studies (with reference to article 9.2.j of the EU general data protection regulation 2016/679 (GDPR)), according to the San Raffaele Hospital internal policy (IOG075/2016).

### 2.3. Statistical Analysis

A two tailed, unequal variances t-test was used to compare 25-(OH)D plasma levels in the SARS-CoV-2 positive and negative groups. A *p*-value < 0.05 was considered for statistical significance. The Kolmogorov–Smirnov Test (*p*-value > 0.05) was used to verify the parametric distribution of the data. For the weekly averaged 25-(OH)D values, we applied the Mann–Whitney test, considering the null hypothesis of no significant difference between the sex and age groups. No overlap between the confidence intervals is consistent with a statistically significant difference.

## 3. Results

### 3.1. Lockdown Effect

In order to verify whether the strict lockdown strategy implemented by the Italian government between 8 March and 18 May 2020 had some effect on the population’s 25-(OH)D levels, we first compared the years 2019 and 2020 in terms of the amount of UV (UVDVC) that reached the Milan area between the 1 January and the 21 August (Figure 1). Figure 1 shows that the UV radiation, suitable for vitamin D biosynthesis, that reached the Milan area in the first 8 months of the 2020, did not differ significantly from that of 2019.

Figure 2 shows the comparison of the weekly averaged 25-(OH)D levels, recorded over two years, stratified by age and gender. Since the effect of UV radiation on 25-(OH)D levels is typically observed after two months [27] any effect caused by the COVID-19-related lockdown would have been observed approximately between May and July 2020. Figure 2 shows that during such a period, the 25-(OH)D levels were significantly higher (confidence interval (C.I.) not overlapping) in 2020 than in 2019. This effect is more evident when considering females and older individuals, whereas it is almost absent in the male group aged 21–45 (Figure 2). Figure 2 also shows that during March–April 2020, there was a marked drop of the 25-(OH)D levels in the oldest group.

When comparing the 2020 25-(OH)D levels among the sub-cohorts stratified for age and gender (Figure 3), it is evident that the significant (*p*-value < 0.05) drop in 25-(OH)D concentrations observed in March is mainly due to measurements from older females which, in contrast, experience higher vitamin D levels during the remaining part of the graph.

### 3.2. First Wave vs. Second Wave

The association between SARS-CoV-2 positivity and 25-(OH)D status in the Milan area during the first COVID-19 Italian wave has been previously published by this group [21] and the results are summarized in Table 1.

During the second Italian COVID-19 wave, approximately framed between 1 October and 30 November, 89,666 SARS-CoV-2 RT-PCR tests (33,641 positives and 56,025 negatives) were performed, of which 1373 (188 positives and 1185 negatives) were associated with at least one 25-(OH)D measurement between 1 June 2020 and 30 November 2020 (Table 2). During this period, 27,684 25-(OH)D measurements were performed in the same hospital. Although information about the origin of vitamin D requests were not available, only 5% were from hospitalized patients. We might assume that the remaining 95%, from outpatients, were mainly associated to routine vitamin D monitor requests. Furthermore, information about patients’ vitamin D supplementation were also lacking. In contrast to the first wave, according to the Fisher’s test, no significant differences were observed in the male/female distribution for both SARS-CoV-2 positive and negative groups. The averaged time interval between the patients’ swab test and their corresponding 25-(OH)D determination were 31 ± 56 days. Averaged 25-(OH)D values for SARS-CoV-2 positive (SARS-CoV-2 +) and negative (SARS-CoV-2 **–**) individuals were 25.1 ± 13.2 and 26.7 ± 13.3 ng/mL, respectively. The two groups showed statistically similar 25-(OH)D values (*p*-value 0.125), whereas both were significantly lower than the general population (*p*-value < 0.005; Table 2). When compared to the first wave, both positive and negative groups showed significantly higher 25-(OH)D values during the second wave.

The 1373 individuals with either a positive or negative swab test were further stratified, according to their age, in three different groups. SARS-CoV-2 + and SARS-CoV-2 − groups were compared to each other as well as with the 25-(OH)D concentrations measured, in the same period in the general population of Milan (Table 3). No statistically significant differences were observed between the negative and positive groups with the exception of females older than 65 years of age where the SARS-CoV-2 infected individuals had vitamin D levels significantly lower (*p*-value < 0.001) than those recorded in the negative group. In contrast, when the negative and positive groups were compared to the averaged vitamin D concentration of the general population, their levels were statistically different, with the only exception of SARS-CoV-2 + males younger than 46 years (*p*-value 0.6174), SARS-CoV-2 − females younger than 46 years (*p*-value 0.0818) and SARS-CoV-2 + females aged 46–65 (*p*-value 0.4451). All the individuals older than 65 years of age, with either a negative or positive swab test, had f25-(OH)D concentrations significantly lower than the general population.

Table 3 also shows that, in the general population, the averaged 25-(OH)D levels were very similar and below the sufficiency level of 30 ng/mL for males (any age) and younger females (< 46 years) whereas, despite the recent passage through warm season, only females older than 46 have, on average, sufficient 25-(OH)D concentrations.

According to previous studies [27] at this latitude, during summer, 10–20 min of sun exposure (25% of the body surface) are sufficient to synthesize an optimal amount of vitamin D. Thus, sun exposure alone cannot explain the low 25-(OH)D levels observed in this study. We might speculate that other factors like the current lifestyle (working hours, dress code, etc.) might be the reason of the high percentage of vitamin D insufficiency recorded during the studied period.

## 4. Discussion

As regards the association between SARS-CoV-2 infection and COVID-19 mortality rates with sunlight exposure, one (but not the main) factor involved in endogenous vitamin D biosynthesis [27] has been almost neglected. In general, the few published studies indicate the existence of a fair but significant association between the diseases outbreak and latitude, although they are limited to the first few months of the pandemic [30,31]. A more updated analysis has shown a possible correlation between COVID-19 cases every million of inhabitants and 25-(OH)D levels, but not with COVID-19-related death rates [2]. Rhodes and colleagues studied the situation in 117 countries with a population above one million and recorded more than 150 COVID-19 cases. The analysis has identified a threshold at 28° north (the latitude of the capital city was considered). After adjustment for age, the authors estimated a mortality rate increase of 13.7% (95% CI 7.4%–20.3%) for each 1% increase in % ≥ 65 years of age, which was highly significant. Furthermore, while population density was not significantly associated with mortality, the authors found a significant association with latitude with an estimated 4.4% (95% CI 0.4%–8.5%) increase in mortality for each 1° further north [32]. This study supports the findings about the link between latitude and COVID-19 mortality reported within the USA [33]. Similarly, Moozhipurath et al. [34] identified a negative association between UV intensity (UVI) and COVID-19 deaths. A fixed-effect log-linear regression model applied to data from 152 countries over 108 days found that a permanent unit increase in UVI associates with 1.2% decline in the daily growth rates of cumulative COVID-19 deaths and 1.0% decline in the case-fatality rate (CFR) daily growth. Thereby, a 12% reduction in terms of the daily growth rates of cumulative COVID-19 deaths and 38% reduction for CFR was described [34]. These findings were confirmed by Tang et al. who indicated that the average percent positive of four common coronaviruses (CoVHKU1, CoVNL63, CoVOC43, and CoV229E) and SARS-CoV-2, in a month, based on US geographical distribution along the latitude, negatively correlated with sunlight UV radiation dose [35]. During the period 1 January–30 August, no significant differences in UVDVC were found between 2019 and 2020. In particular, since the effects of UV radiation on 25-(OH)D levels are typically observed after two months [27], any impact of the pandemic-related lockdown took place approximately between May and July 2020, as the UVDVC profiles overlap (Figure 1). However, when vitamin D status is considered, it becomes evident that in such a period, the 25-(OH)D concentrations were significantly higher in 2020 than in 2019. This effect is more evident when considering females and older individuals, whereas it is almost absent in the male group aged 21–45 (Figure 2). Although speculative, it is possible to hypothesize that, during the lockdown period, in order to limit the negative effects of confinement, a general suggestion to supplementation with vitamin D was given by a physician to the elderly. The marked drop in 25-(OH)D concentrations observed between March and April 2020, was likely a consequence of the strict directives imposing the priority for clinical examination (as well as admission to emergency room) to those patients with pronounced acute respiratory syndrome symptoms. Consequently, during the lockdown period of the first pandemic wave, only admitted sick patients were tested for vitamin D because they were most likely to be in a hypovitaminosis D status. This matches with the existing evidence about the associations between vitamin D status and the susceptibility and severity of respiratory tract infections [19,20]. The drop in 25-(OH)D was mainly due to measurements from elderly females (e.g., those admitted to the hospital) which, in contrast, experienced higher vitamin D levels during both the pre- and post-lockdown timeframes. This latter aspect is likely related to the common vitamin D supplementation, in this age group, for osteoporosis prevention (Figure 3). As a further confirmation of this hypothesis, the first wave lockdown corresponds to the period in which vitamin D values are lacking among the young subjects (0–20 years), as reported in Figure 2 and Figure 3, who were only marginally involved in the epidemic [36].

The serum 25-(OH)D concentrations measured during the second wave in SARS-CoV-2 + and SARS-CoV-2 – subjects were lower than those measured in the general population (Table 2), but in SARS-CoV-2 + males and females younger than 46 years and SARS-CoV-2 + females aged 46–65 years. All the individuals older than 65 years of age, regardless the SARS-CoV-2 status, had 25-(OH)D levels significantly lower than those of the general population. Furthermore, despite the warm season, only females older than 46 years experienced sufficient vitamin D levels and this was likely the consequence of the strongly recommended vitamin D supplementation in post-menopausal women.

It must be noted that the first and second wave represents two distinct phases of the pandemic. During the first wave there was a shortage of swab tests that, as mentioned above, limited this diagnostic tool to individuals showing pronounced acute respiratory syndrome symptoms. In contrast, during the second wave, the swab test was also performed on COVID-19-close-contacts which, in some cases, were simply healthy individuals. Such a difference is reflected in the swab test of the male/female ratio which was significantly higher than 1 in the first wave (men are more likely to contract SARS-CoV-2) whereas no significant differences were observed in the second wave. In contrast, both the first and second waves showed that the averaged 25-(OH)D levels of those RT-PCR tested for SARS-CoV-2 were significantly lower than those of the general population. Such data suggest that adequate vitamin D levels have a protective effect against both SARS-CoV-2 (positive group) and general respiratory tract infections (negative group). However, a generalization linking the amount of UVR (latitude) and the number of SARS-CoV-2 cases might not be appropriate. It is well known that because of the current lifestyle (people spend little time outdoor and current dress codes imply covering the majority of the skin, even in the warm season) the UVR only partially influence 25-(OH)D levels [23,27]. In addition to the reliability and the ease of retrieval of UVR data from the public database, this method would be not the most appropriate one. On the contrary, the use of a dosimeter would have given a more accurate measurement of UV exposure. Furthermore, in a pandemic scenario, infections are largely influenced by a country’s population density as well as the local lockdown strategies which might differ from country to country [37]. Based on the updated data (Appendix A), the association between SARS-CoV-2 infected people and 25-(OH)D concentrations in different countries is not statistically significant (*p*-value 0.4557), as well as for the association between COVID-19-related deaths and 25-(OH)D (*p*-value 0.3782).

Although it suffers from the impossibility to define a causal-effect link between vitamin D status and SARS-CoV-2 infection risk, this study has the advantage of a wide cohort of subjects analyzed with a geographically limited but densely populated area that was the epicenter of the first pandemic wave. This aspect is of particular relevance for the scientific community engaged in the study of SARS-CoV-2. The Milan area resembles many densely populated areas where containing the pandemic is more difficult. Furthermore, the large cohort of subjects is a reliable cross-section of the general population and our results may give a consistent address to the future studies highlighting more specific aspects of the impact of vitamin D status in the evolution of the infection.

## 5. Conclusions

While several retrospective studies have described a correlation between hypovitaminosis D and SARS-CoV-2 infection and COVID-19 outcomes, many others did not. Nonetheless, most of COVID-19 patients are elderly, and often experience a poor vitamin D status and this represents a major confounding factor in the definition of an association between 25-(OH)D concentrations and COVID-19. In our study, we evidenced the lack of a direct association between solar exposure, 25-(OH)D status and SARS-CoV-2 infection. Furthermore, the stringent confinement implemented during the first wave lockdown did not have any effect on 25-(OH)D status, although some peculiar time-restricted profile behaviors can be deduced possibly due to vitamin D supplementation habits and features of those patients who presented to the hospital during the most dramatic period.

## Figures and Tables

**Figure 1 biology-10-00237-f001:**
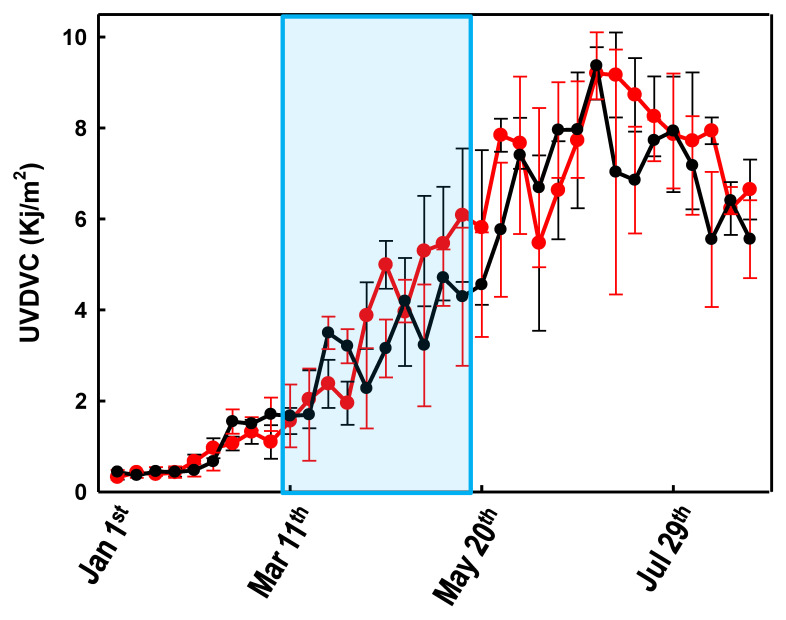
Variations of the weekly averaged (cloud-modified vitamin D UV dose) UVDVC between 1 January and 30 August 2019 (black line) and 2020 (red line). Error bars represent the confidence intervals. The light blue area represents the Italian lockdown period (8 March and 18 May 2020).

**Figure 2 biology-10-00237-f002:**
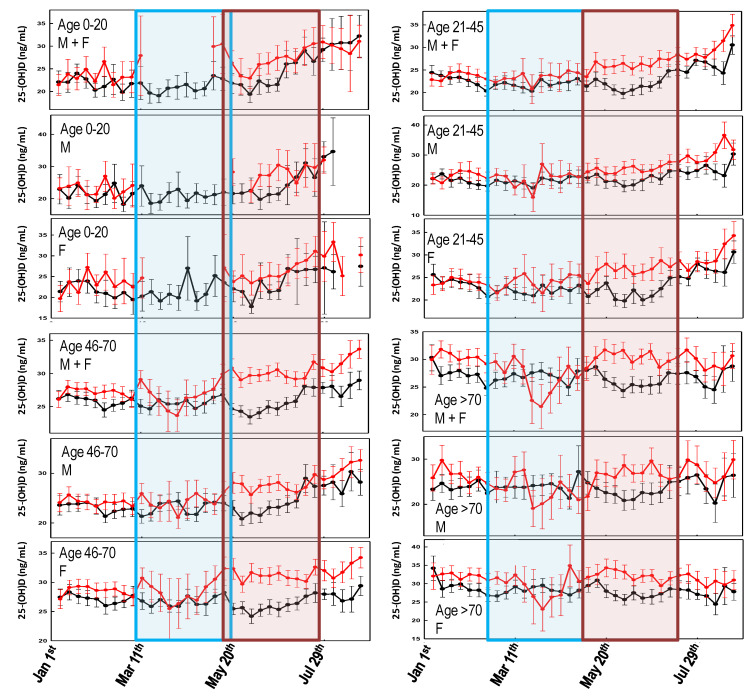
Variability of the weekly averaged 25-(OH)D values for the different age groups and genders, between 1 January and 30 August 2019 (black line) and 2020 (red line). Error bars represent the confidence intervals. The light blue area represents the Italian lockdown period (8 March and 18 May 2020). The light red area represents the expected timing of the lockdown effect on 25-(OH)D levels. No overlap between the confidence intervals is consistent with a statistically significant difference.

**Figure 3 biology-10-00237-f003:**
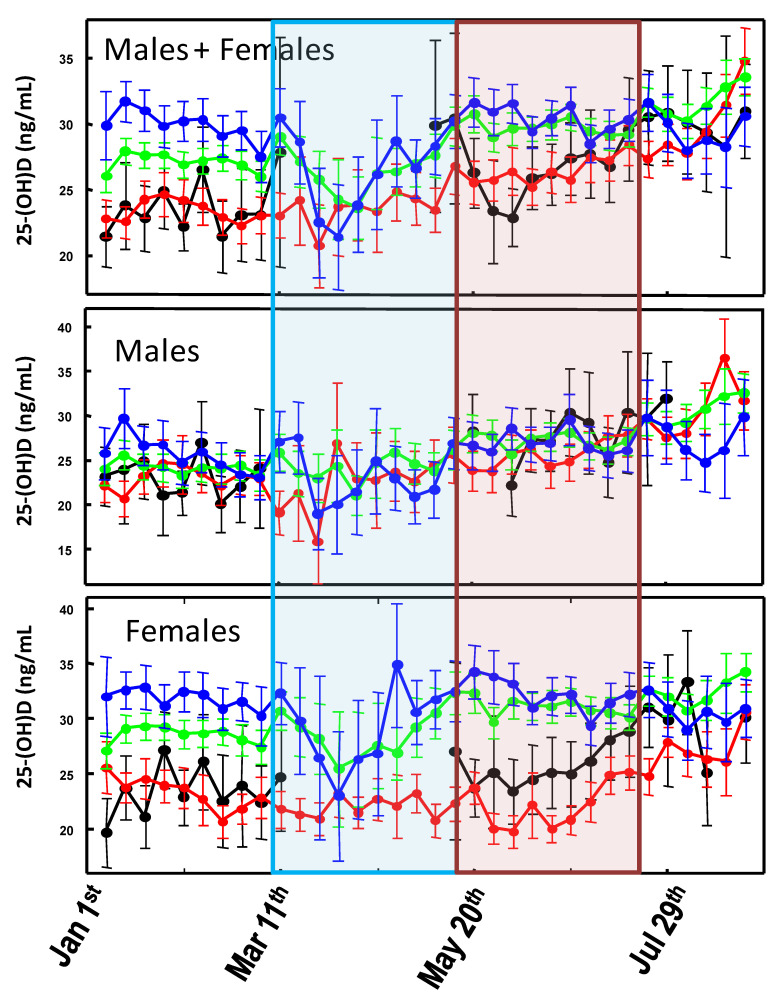
Variability of the weekly averaged 25-(OH)D values measured between 1 January and 30 August 2020 for different age groups: 0–20 (black line); 21–45 (red line); 46–70 (green line); >70 (blue line). Error bars represent the confidence intervals. Weekly averaged values with less than 15 measurements were omitted from the graph. The light blue area represents the Italian lockdown period (8 March and 18 May 2020). The light red area represents the expected timing of the lockdown effect on 25-(OH)D levels.

**Table 1 biology-10-00237-t001:** Demographic characteristics of the 347 subjects RT-PCR tested between 1 March and 30 April for which 25-(OH)D levels were also recorded. *p*-values for age and 25-(OH)D were calculated from the “Total” column values. Age (years) and 25-(OH)D concentrations (ng/mL) are reported as averaged values ± SD.

	SARS-CoV-2 +			SARS-CoV-2 −			
	Males	Females	Total	Males	Females	Total	*p*-Value
N	83 (64.8%)	45 (35.2%)	128 (100%)	107 (48.9%)	112 (51.1%)	219 (100%)	<0.05 *
Age (years)	62.7 ± 14.2	69.3 ± 15.6	65.0 ± 15.0	62.8 ± 19.5	54.3 ± 20.1	58.7 ± 20.2	<0.05
25-(OH)D (ng/mL)	19.7 ± 13.5	25.8 ± 19.6	21.8 ± 16.1	23.1 ± 15.0	22.6 ± 13.1	22.8 ± 14.0	0.39

* According to the Fisher’s test, only the positive group has a male/female ratio statistically different from 1.

**Table 2 biology-10-00237-t002:** Demographic characteristics and 25-(OH)D levels of the patients involved in the study. SARS-CoV-2 positive (SARS-CoV-2 +) and SARS-CoV-2 negative (SARS-CoV-2 **−**) groups refers to patients RT-PCR tested between 1 October and 30 November 2020. The whole population group refers to patients tested for vitamin D between 1 June and 30 November 2020. Age (years) and 25-(OH)D concentrations (ng/mL) are reported as averaged values ± SD. # indicates a statistically significant difference vs. the gender-matched total population (###; *p*-value < 0.001).

	**SARS-CoV-2 +**		
	**Males**	**Females**	**Total**
N	101 (53.7%)	87 (46.3%)	188 (100%)
Age (years)	58.9 ± 21.6	61.4 ± 18.8	60.0 ± 20.3
25-(OH)D (ng/mL)	23.7 ± 11.5 ^###^	26.7 ± 14.7	25.1 ± 13.2
	**SARS-CoV-2 −**		
	**Males**	**Females**	**Total**
N	547 (46.1%)	638 (53.9%)	1185 (100%)
Age (years)	55.5 ± 23.9	54.0 ± 22.3	54.7 ± 23.0
25-(OH)D (ng/mL)	24.9 ± 12.9 ^###^	28.2 ± 13.5	26.7 ± 13.3
	**Whole population**		
	**Males**	**Females**	**Total**
N	9164 (33.1%)	18,520 (66.9%)	27,684 (100%)
Age (years)	56.6 ± 19.6	59.0 ± 18.3	58.2 ± 18.8
25-(OH)D (ng/mL)	29.0 ± 12.0	31.7 ± 13.5	30.8 ± 13.1

**Table 3 biology-10-00237-t003:** Comparison between the averaged 25-(OH)D levels, stratified by age groups. SARS-CoV-2 + and SARS-CoV-2 − groups refers to patients RT-PCR tested between 1 October and 30 November 2020. Total refers to the patients tested for 25-(OH)D between 1 June and 30 November 2020. The 25-(OH)D concentrations (ng/mL) are reported as averaged values ± SD; in brackets, the number of patients. § indicates a statistically significant difference between SARS-CoV-2 + and the gender-matched SARS-CoV-2 − individuals; # indicates a statistically significant vs. the gender-matched total population (§: *p*-value < 0.05; ##: *p*-value < 0.01; ###: *p*-value < 0.001).

		<46	46–65	>65
		Females		
25-(OH)D (ng/mL)	SARS-CoV-2 +	* (13)	30.4 ± 14.8 (32)	22.2 ± 13.5 (42) ^§ ###^
	SARS-CoV-2 –	27.7 ± 11.3 (200)	29.5 ± 14.2 (201) ^##^	27.6 ± 14.4 (237) ^###^
	Total	29.2 ± 11.9 (3728)	32.2 ± 13.3 (7490)	32.7 ± 15.0 (7300)
		**Males**		
25-(OH)D (ng/mL)	SARS-CoV-2 +	28.5 ± 10.6 (21)	23.1 ± 12.8 (34) ^##^	22.0 ± 10.5 (46) ^##^
	SARS-CoV-2 −	26.5 ± 10.3 (161) ^##^	26.2 ± 13.0 (147) ^###^	23.1 ± 14.2 (239) ^###^
	Total	29.5 ± 11.6 (2335)	29.4 ± 11.4 (3572)	28.3 ± 13.8 (3257)

* Age groups including less than 15 patients were omitted.

## Data Availability

The data presented in this study are openly available in Zenodo at doi: 10.5281/zenodo.4462474.

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
