# Peer review of "Changes in 25-(OH) Vitamin D Levels during the SARS-CoV-2 Outbreak: Lockdown-Related Effects and First-to-Second Wave Difference—An Observational Study from Northern Italy"

_biology, 2021, doi:10.3390/biology10030237_

Round 1
Reviewer 1 Report
Ferrari et al report a comprehensive analysis of 25OHD throughout the COVID-19 pandemic in the Milan area in Italy, hoping to assess both the effects of lockdown on 25OHD levels and also a potential association of such levels to predisposition to COVID-19. Overall the manuscript is robust and well written, with impactful results. However, it feels as if the authors expected a different result and wrote this manuscript with a heavy heart, as they tend to keep “apologizing” for their null findings, which, in reality, are bona fide and should be reported as such. In my view, a few adjustments are necessary, as detailed below:
1) The text of the “simple summary” on page 1 stroke me as too aggrandizing. In fact, the tone of the summary differs from that of the text and abstract. Examples:
- “vitamin D has several functions into the body and, besides its role in bone health, its optimal status is related with a reduced risk of respiratory tract infections” – this is a bit of extrapolation. As the authors themselves state later, “vitamin D is SUPPOSED to be MORE OR LESS directly involved in a plethora of physiological mechanisms”. Therefore, this first sentence of the summary needs adjustment;
- the final sentence is even more misleading: “However, the existence of indirect relationship and a protective role of an optimal vitamin D status on general respiratory tract infections was observed.” This does not match the observed results. The authors should be very clear about what they found, without getting carried away by what they thought they would have found or what they would like to believe.
2) For consistency, I suggest using “25OHD levels” instead of “vitamin D levels” throughout the text. In the summary and in 2.2 (clinical data) “vitamin D levels” is used, for example.
3) Page 2, Introduction, end of 2nd paragraph: the authors should spell out what the “and confirmed” means - to my knowledge, the only hard evidence of VitD supplementation having a role in prevention of airway infection is the one metaanalysis by Martineau (one study amongst several RCTs and metaanalysis, indeed). The references provided are for review studies and not original data. Please describe the original data and refer to it. And I would avoid “confirmed” as a word in this scenario here. Nothing is much confirmed about VitD extraskeletal actions.
4) Page 3, 2.2. Clinical data:
- Are the three cohorts (2019, 2020 and second wave) balanced for presentation of male/female and age groups? Please provide a statistical analysis of that.
- Please provide interassay and intraassay coefficients of variation for the determination of 25OHD in your center – these are usually high (El-Fulleihan et al, JBMR 2015) and certainly impact on the appreciation of differences in 25OHD between 20 and 30 ng/ml
5) Page 9, 4th paragraph: “Our analysis, however, cannot explain whether vitamin D may have a protective effect on SARS-CoV-2 infection in the elderly or not and if young subjects with hypo-vitaminosis D were mainly asymptomatic and, hence, were not tested for SARS-CoV-2.” This is very speculative, and, again, reads as the authors are apologizing for their findings. Please rephrase if you want to state this limitation.
5) Page 10, Conclusions: the first three sentences are NOT conclusions of this paper, substantiated by its results, but rather they reflect the authors views of the available literature. Their evidence-based conclusion starts on the fourth sentence (“Our study…”). Therefore, these first 3 sentences do not belong in Conclusions, and, if kept, should be incorporated somewhere in the discussion.
Author Response
Please, refer to the attached file.

Reviewer 2 Report
This observational study from Northern Italy, investigates the changes in 25-(OH)D levels in different age groups during the first lockdown period in relation with the recorder sunlight exposure, compared to the same period during the previous year. This study also compares the differences of 25-(OH)D levels between the first and second waves in association with positivity to SARS-CoV-2 nasopharyngeal swab test.
In general, the manuscript is well designed, and the results are well presented.
There are some minor issues (see pdf attached) along the manuscript that need to be addressed to consider this manuscript for publication.

Author Response
The authors are grateful to this Reviewer for the precious and punctual comments. The authors have taken into consideration all the comments and where possible have modified the text. Otherwise, a justification has been provided.
The comments are reported as responses in the pdf attached.

Reviewer 3 Report
The manuscript has investigated the changes in the levels of vitamin D during lockdown and its correlation with CIVID-19 cases. The conclusion is supported by the data provided, however, some changes are needed
- Please check for the English language as often sentence formation is confusion (e.g. Taken together, our data suggest that vitamin D levels are directly related neither with sun exposure nor with SARS-CoV-2 infection; barrier cells, involved in the identification of pathogens express the; severity of COVID-19 outcomes, included death [2]. etc. )
- The discussion section needs to be rewritten. The first paragraph has summarized previous literature; the second paragraph discuss the aim of this study; and third paragraphs has described the results again. The text looks redundant.
- The authors have mentioned "Taken together, our data suggest that vitamin D levels are directly related neither with sun exposure", please explain the reason for this in the discussion. The negative and positive findings should be discussed in the discussion section and not just summarizing the results again.
- The authors mentioned that this article adds a large cohort study to the existing data. The authors should discuss the advantage of this study and how this study will help to the scientific community or physicians working on COVID?
Author Response
Please, refer to the attached file
